# Optimal Energy-Storage Configuration for Microgrids Based on SOH Estimation and Deep Q-Network

**DOI:** 10.3390/e24050630

**Published:** 2022-04-29

**Authors:** Shuai Chen, Jinglin Li, Chengpeng Jiang, Wendong Xiao

**Affiliations:** 1School of Automation and Electrical Engineering, University of Science and Technology Beijing, Beijing 100083, China; B20150284@xs.ustb.edu.cn (S.C.); B20190280@xs.ustb.edu.cn (J.L.); B20160276@xs.ustb.edu.cn (C.J.); 2Key Laboratory of Knowledge Automation for Industrial Processes of Ministry of Education, School of Automation and Electrical Engineering, University of Science and Technology Beijing, Beijing 100083, China

**Keywords:** microgrid, deep Q-network, state of health, electric/thermal hybrid energy storage, optimal configuration

## Abstract

Energy storage is an important adjustment method to improve the economy and reliability of a power system. Due to the complexity of the coupling relationship of elements such as the power source, load, and energy storage in the microgrid, there are problems of insufficient performance in terms of economic operation and efficient dispatching. In view of this, this paper proposes an energy storage configuration optimization model based on reinforcement learning and battery state of health assessment. Firstly, a quantitative assessment of battery health life loss based on deep learning was performed. Secondly, on the basis of considering comprehensive energy complementarity, a two-layer optimal configuration model was designed to optimize the capacity configuration and dispatch operation. Finally, the feasibility of the proposed method in microgrid energy storage planning and operation was verified by experimentation. By integrating reinforcement learning and traditional optimization methods, the proposed method did not rely on the accurate prediction of the power supply and load and can make decisions based only on the real-time information of the microgrid. In this paper, the advantages and disadvantages of the proposed method and existing methods were analyzed, and the results show that the proposed method can effectively improve the performance of dynamic planning for energy storage in microgrids.

## 1. Introduction

A microgrid is a small power generation and distribution system composed of distributed power sources, energy storage devices, energy conversion devices, loads, monitoring and protection devices, etc. Its purpose is to realize the flexible and efficient application of distributed power, and to solve the problem of the grid connection of large-scale and diverse distributed energy. Microgrids fully promote the large-scale access of distributed energy and new energy, and can provide users with a reliable supply of multiple energy sources [1,2]. As an important part of the smart grid, the microgrid can realize the integrated operation of the internal power supply and the electricity load and can coordinate with the main grid, smoothly connect to the public grid, or operate independently. The flexible scheduling capability of the microgrid can meet the user’s requirements for power quality, power reliability, and safety [3].

Energy storage is an important part of microgrids needed to realize multi-energy coupling, which can realize the transfer of energy across time periods. It can not only maintain the balance of energy supply and demand, smooth the output of new energy, and restrain the fluctuation of renewable energy power, but also play an important role in improving power quality, safe and reliable operation, and cost-effective coordination [4]. Compared with other storage technologies, battery energy storage has the advantages of a small footprint, no geographical restrictions, and a fast response speed. Lithium batteries are widely used due to their advantages of high energy density, low self-discharge, fast charge and discharge, and no memory effect. The energy storage battery does not directly generate operating costs during operation, but it is expensive. At the same time, its battery life is affected by the number of charging and discharging events, which limits its large-scale promotion and application [5]. Therefore, the optimal energy configuration of microgrids is becoming an important research direction at present. In practical application, the life of energy storage is continuously lost with running time. Compared with battery energy storage, thermal energy storage equipment can operate stably in the planning cycle with low cost. In many studies, the optimal configuration of hybrid energy storage mainly considers the impact of battery life loss and has achieved a good performance.

The energy optimization configuration considering the life of the energy storage battery is a nonlinear multi-stage dynamic programming problem. It is not appropriate to introduce an overly complex energy storage model, and it needs to be able to truly reflect the impact of the life of the energy storage battery. The optimization models of various energy storage capacities have been widely studied. Some hierarchical optimization planning methods for hybrid energy storage capacity have been proposed to solve the shortage of independent planning of electric/thermal energy storage [6,7,8]. The article [9] considers the uncertainty of multi-load demand and proposes a collaborative optimization method for electric/thermal hybrid energy storage in the planning and operation stages. The article [10] proposes an optimal configuration method of electric/thermal hybrid energy storage based on the energy balance from the perspective of user-side electric energy substitution. Some multi-energy optimal scheduling strategies are proposed, which have good economics in the local area [11,12,13,14]. However, these algorithms do not consider the operation performance and economic cost in the scenario of interconnection with the public grid. The above studies contain a relatively insufficient consideration of battery life loss in microgrid energy storage configuration.

The research results of the article [15] show that energy storage planning without considering battery life loss makes it difficult to achieve optimal performance and will face the risk of overestimating the investment benefits of energy storage. At present, some studies have used simple constraints to limit the influencing factors of life loss, but have not given a quantitative evaluation and measurement of battery life loss. The accurate measurement method of battery life loss based on data analysis does not need to consider the deterioration process and principle of the battery and can be quickly modeled by using a neural network. Deep learning can extract data features from a high-dimensional state space, which can simplify the modeling of many complex scenarios. However, the neural network parameters used by some methods are too complex, resulting in a large computational load in the planning stage [16,17,18,19]. Therefore, in-depth research is needed to adapt the battery life model to actual scenarios.

Aiming at the problem of the optimal configuration of energy storage in multi-energy microgrids, this paper constructs a model of battery life loss in multi-energy microgrid planning problems and reflects the long-term impact of battery life loss in the form of battery replacement costs. At the same time, the economic optimization of the multi-energy microgrid including energy storage is considered in the planning stage, and the operation of the microgrid and the economy of energy storage are taken into account. Based on the above analysis, this paper designs a battery life estimation model based on a lightweight neural network. At the same time, considering the energy storage battery life and the economy of energy consumption in the microgrid, this paper designs a two-layer optimization model and forms a microgrid energy storage configuration strategy that takes into account the battery life loss.

The paper is arranged as follows: Section 2 gives a brief introduction to the microgrid energy storage structure. Section 3 describes the battery life loss estimation based on lightweight neural networks. In Section 4, the energy storage optimization model based on battery life loss prediction is constructed. Section 5 shows the experimental results. Finally, conclusions are given in Section 6.

## 2. Review of Microgrid Energy Storage Structure

A microgrid is an important part of a smart grid, which can ensure the stable operation of a power system. It is an autonomous system that can operate either with an external grid or in isolation. The operation and management of microgrids involves the diversity and complexity of the power sources and loads. At the same time, the role of microgrids in smart cities, smart parks, and smart homes is becoming more and more important, and their operation modes are increasingly complex. A typical microgrid mainly includes units of energy production, energy conversion, energy storage, energy transmission, and energy consumption [20,21]. As shown in Figure 1, a microgrid consists of photovoltaic power generation, energy storage batteries, combined heat and power (CHP) units, heat pumps, and loads.

## 3. Battery Life Loss Estimation Based on Lightweight Neural Networks

Lithium batteries continue to deteriorate with the increase in the number of charging/discharging events, and the remaining capacity of the battery gradually decays. After the deterioration reaches a certain level, the battery will become unusable. The National Aeronautics and Space Administration (NASA) analyzed the charging and discharging data of lithium batteries under different working conditions [22]. Figure 2 shows that the residual capacity of the fifth, sixth, and seventh batteries decreases continuously with the increase in the charging/discharging events.

In order to effectively measure the deterioration degree of the battery, the state of health (SOH) of the battery was used to describe the deterioration degree of the current state relative to the original state. Its expression is represented by
(1)$$\mathrm{SOH}=\frac{c_{now}}{c_{0}}\times 100$$
where $c_{now}$ and $c_{0}$ are the current full charge capacity and the nominal capacity of the lithium battery, respectively. When SOH decays to about 80%, the battery is considered to be at the end of life.

At present, the SOH estimation methods of lithium batteries mainly include direct measurement methods, model-based methods, and data-driven methods [23,24,25]. The data-driven method analyzes historical data for battery SOH estimation through a fully connected neural network and CNN model, and it does not require much prior knowledge about the battery mechanism. However, the neural network architectures of these methods are complex, the number of parameters is large, and the computational efficiency is low, which cannot be performed on a large scale or on resource-constrained embedded battery management devices [26,27,28]. Therefore, we propose to use the lightweight convolutional neural network MobileNetV3 to estimate the lithium battery SOH.

### 3.1. MobileNetV3 Model

With the continuous increase in intelligent applications, the parameter quantity and operation speed of deep neural networks (DNNs) have become the key factors for practical application. Lightweight neural networks have gradually become a current research hotspot. The MobileNet series is a very important lightweight network family proposed by Google. MobileNetV3 adopts separable convolution technology, which reduces the complexity and parameter amount of neural networks and can be well applied to scenarios with limited storage and power consumption [29,30,31]. Meanwhile, the model introduces a channel attention mechanism (squeeze-and-excite, SE) to adjust the weight of each channel, which improves the training and inference performance of the neural network.

MobileNetV1 uses depth to separate convolution to build a lightweight network. MobileNetV2 proposes an innovative inverted residual with a linear bottleneck unit. Although the number of network layers has increased, the overall network accuracy and speed have improved. MobileNetV3 was optimized on the basis of MobileNetV1 and MobileNetV2, using separable convolution, an inverted residual structure, and a lightweight attention model and the performance and speed are even better. The models of MobileNetV3 are MobileNetV3-Large and MobileNetV3-Small, and the number of neural network layers is 20 and 16 respectively, which can match different resource conditions. Compared with MobilenetV2, the ImageNet classification accuracy of MobilenetV3-Large and MobilenetV3-Small is improved by about 4.6% and 3.2%, respectively, and the computation time is reduced by 5% and 15%, respectively.

The first layer of the MobileNetV3-Small neural network is a two-dimensional convolutional layer. The middle layer is a 11-layer residual neural network. The last part comprises the convolutional layer, the pooling layer, and two convolutional layers. In this neural network model, the h-swish activation function is used to replace the swish function, which further reduces the amount of computation. The mathematical expressions of swish and h-swish are as follows
(2)$$\mathrm{swish}\left[x\right]=x\cdot \sigma \left(x\right),$$
(3)$$h-\mathrm{swish}\left[x\right]=x\frac{\mathrm{ReLU}6\left(x+3\right)}{6},$$
where *x* is the input value. Considering that the calculation of the sigmoid function takes a long time, MobileNetV3-Small uses ReLU6(*x* + 3)/6 instead of the sigmoid function, which can improve the efficiency by about 15%.

### 3.2. SOH Prediction Based on Modified MobileNetV3-Small Model

The MobileNetV3-Small model is mainly used to process two-dimensional data. In this paper, the three factors of current, voltage, and temperature of lithium battery charging are composed of two-dimensional data (3 × *N*) according to time series, and *N* is the number of parameters collected during the charging process. According to practical experience, the sampling times were set to 300, and the size of the two-dimensional data was 3 × 300. Since the size of the input data of the MobileNetV3-Small model is 224 × 224, we needed to adjust the MobileNetV3-Small to suit the needs of SOH prediction. The neural network structure of the modified MobileNetV3-Small (MobileNetV3-Small-M) is shown in Table 1. Meanwhile, MobileNetV3-Small is usually used to solve the problem of image classification. The SOH prediction of batteries is a regression problem. Therefore, the last layer of the neural network needed to adjust the output data to one-dimensional data through a 1 × 2 convolution operation.

In order to meet the needs of lightweight neural network training, adaptive momentum estimation algorithm was proposed to improve the training performance of neural networks [32]. Adam is a first-order optimization algorithm that can replace the traditional stochastic gradient descent process. Its core is to iteratively update neural network weights based on training data. Meanwhile, in order to solve the slow gradient and gradient dispersion in traditional gradient descent calculation, the cross entropy loss function is proposed as the loss function for neural network training.

Cross entropy can measure the degree of difference between two different probability distributions in the same random variable and can describe the difference between the true probability distribution and the predicted probability distribution. The smaller the value of the cross-entropy, the better the model predicts. The cross entropy loss function can be defined as
(4)$$C=-y\log\left(\hat{y}\right)-\left(1-y\right)\log\left(1-\hat{y}\right),$$
where *y* and $\hat{y}$ are the true value and predicted value, respectively.

Based on the operating data of the microgrid, the improved lightweight convolutional neural network can not only reduce the requirements of the computing resource, but also improve the training speed of the neural network. It provides a good evaluation method for the life loss of the energy storage battery.

### 3.3. Quantification of Battery Life Loss

SOH is an important index to evaluate battery performance. Each discharging process is a gradual consumption of the battery ’s nominal capacity $c_{0}$, so the battery life loss cost of the *i*-th discharging process $c_{i}^{loss}$ can be defined as
(5)$$c_{i}^{loss}=\frac{{\mathrm{SOH}}_{i-1}-{\mathrm{SOH}}_{i}}{80\%}C^{cap},$$
where *C^cap^* is the initial battery investment cost.

## 4. Comprehensive Energy Storage Optimization Considering Battery Life Loss

Due to the diversification of energy demands of various loads, the coupling and complementation of various energy sources in the microgrid can effectively improve the efficiency of energy utilization. Compared with battery energy storage, the hybrid energy storage system has the characteristics of a low cost and long cycle and can operate stably in microgrids. Simultaneously, it has a good schedulability. Therefore, the impact of battery life loss is mainly considered in the optimal configuration of comprehensive energy storage [33,34,35]. In this paper, the optimization problem of electric/thermal energy storage is considered comprehensively, and a two-layer model with outer layer optimization and inner layer optimization is designed. As shown in Figure 3, a two-layer optimization structure is constructed to solve the problems of battery energy storage capacity planning and system operation optimization. In the two-layer optimization model, the outer layer optimization model is to search the optimum results with the maximum economic performance of the microgrid. The inputs are the economic parameters, system parameters, and the results of the inner layer optimal model, etc. The optimal energy storage battery capacity is used as the output. The inner layer optimization model takes the minimization of the system operating cost as the optimal objective. The inputs are the technical parameters, energy costs, and the results of the outer layer optimization model, etc. The daily operating cost is used as the output.

### 4.1. Outer Layer Optimization

#### 4.1.1. Objective Function

The objective function of the outer optimization model is to minimize the equivalent annual cost during the microgrid operation cycle, that is,
(6)$$\min C=\left(C^{cap}+C^{rep}\right)z^{cr}+C^{om}+C^{fuel}+C^{line},$$
where *C^rep^*, *C^om^*, *C^fuel^*, and *C^line^* are the battery replacement cost, the equipment operation and the maintenance cost, the fuel cost, and the annual power purchase cost, respectively. *z^cr^* is the capital recovery coefficient, and it can be calculated as
(7)$$z^{cr}=\frac{\gamma {\left(1+\gamma \right)}^{l}}{{\left(1+\gamma \right)}^{l}-1},\;\;$$
where *γ* and *l* are the discount rate and the operating period, respectively. The investment cost of battery can be expressed as
*C^cap^* = *Q^es^ω^cap^*, (8)
where *Q^es^* and *ω^cap^* are the initial capacity and unit capacity cost, respectively. The battery replacement cost is expressed as
(9)$$C^{rep}=l\underset{m}{\sum }\underset{d}{\sum }\underset{i}{\sum }c_{m,d,i}^{loss}N_{m,d}-C^{cap},$$
where *m* is the month (*m* = 1, 2, …, 12). *d* = 1, 2, and 3 represent a working day, peak day, and rest day, respectively. *N_m,d_* is the number of typical days in month *m*. $c_{m,d,i}^{loss}$ is the battery life loss cost caused by the *i*-th discharge on typical day *d* of month *m*. The fuel cost, operation cost, and power purchase cost are formulated as
(10)$$C^{fuel}=t\underset{m}{\sum }\underset{d}{\sum }c_{m,d}^{fuel}N_{m,d},$$
(11)$$C^{om}=t\underset{m}{\sum }\underset{d}{\sum }c_{m,d}^{om}N_{m,d},$$
(12)$$C^{line}=t\underset{m}{\sum }\underset{d}{\sum }c_{m,d}^{line}N_{m,d},$$
where $c_{m,d}^{fuel}$, $c_{m,d}^{om}$, and $c_{m,d}^{line}$ are the fuel cost, operation cost, and power purchase cost on the typical day *d* of month *m*, respectively.

#### 4.1.2. Constraint Condition

Limited by the site, the constraints of energy storage investment capacity are expressed as
(13)$$0\leq Q^{es}\leq Q^{\max,es},$$
where *Q*^max,*es*^ is the upper limit of the installed capacity.

### 4.2. Inner Layer Optimization

#### 4.2.1. Objective Function

The inner objective function considers the lowest daily running cost, and its mathematical expression is defined as
(14)$$\min c=c_{m,d}^{es}+c_{m,d}^{fuel}+c_{m,d}^{om}+c_{m,d}^{line},$$
where $c_{m,d}^{es}$ is the battery loss cost on typical day *d* of month *m*, which can be used to optimize the charging/discharging operation of the energy storage battery.

#### 4.2.2. Constraint Condition

*(A)* Power Balance

The mathematical expression of the electrical/thermal power balance constraint can be described as
(15)$$P_{m,d,t}^{line}+P_{m,d,t}^{chp}+P_{m,d,t}^{pv}+P_{m,d,t}^{es,dis}=P_{m,d,t}^{el}+P_{m,d,t}^{es,ch}+P_{m,d,t}^{hp},$$
where *t* represents a certain time. $P_{m,d,t}^{line}$ is the power purchased from the public grid. $P_{m,d,t}^{chp}$ is the power supply of CHP. $P_{m,d,t}^{pv}$ is the power consumed by photovoltaics. $P_{m,d,t}^{es,dis}$, $P_{m,d,t}^{el}$, $P_{m,d,t}^{es,ch}$, and $P_{m,d,t}^{hp}$ are the discharge power, load power, charging power, and heat pump power, respectively.

*(B)* Equipment Operation

The constraint of equipment output is expressed as
(16)$$P^{\min,k}\leq P_{m,d,k,t}^{out}\leq P^{\max,k},$$
where *P*^max,*k*^ and *P*^min,*k*^ are the upper and lower limits of the output of the *k*-th device. $P_{m,d,k,t}^{out}$ is the output of the *k*-th device in the period *t*. *k* = 1 and *k* = 2 represent CHP and the heat pump, respectively.

The power constraint for interconnection with the public grid is expressed as
(17)$$P^{\min,line}\leq P_{m,d,t}^{line}\leq P^{\max,line},$$
where *P*^max,*line*^ and *P*^min,*line*^ are the upper and lower limits of power, respectively.

*(C)* Energy Storage

Under the premise of safe operation, energy storage needs to meet power constraint and SOC (state of charge) constraint.

➢Energy storage battery charging: ${\mathrm{SOC}}_{t}=\left(1-\mu^{es}\right){\mathrm{SOC}}_{t-1}+\frac{P_{t}^{es,ch}\eta^{ech}}{Q^{es}}\Delta t,$➢Energy storage battery discharging: ${\mathrm{SOC}}_{t}=\left(1-\mu^{es}\right){\mathrm{SOC}}_{t-1}-\frac{P_{t}^{es,dis}}{Q^{es}\eta^{edis}}\Delta t$.

In the above formula, SOC*_t_* is the SOC of the battery at time *t*. *μ^es^* is the self-discharge rate of the battery. *η^ech^* and *η^edis^* are the battery charging efficiency and discharging efficiency, respectively. *Q^es^* and Δ*t* are the battery installed capacity and unit operating time, respectively. $P_{t}^{es,ch}$ and $P_{t}^{es,dis}$ are the charging power and discharging power of the battery in the period *t*. At the same time, according to the actual operation of energy storage, some constraints need to be added to improve the effectiveness of the strategy.
(18)$$0\leq P_{t}^{es,ch}\leq \zeta_{t}P^{\max,ch},$$
(19)$$0\leq P_{t}^{es,dis}\leq \zeta_{t}P^{\max,dis},$$
(20)$${\mathrm{SOC}}^{\min}\leq {\mathrm{SOC}}_{t}\leq {\mathrm{SOC}}^{\max},$$
where SOC^max^ and SOC^min^ are the maximum value and the minimum value of SOC, respectively. *ζ_t_* = 1 means it is in energy storage mode. *ζ_t_* = 0 means it is in power consumption mode. The energy storage state at the beginning and the end of the operation cycle needs to satisfy ${\mathrm{SOC}}_{t_{0}}={\mathrm{SOC}}_{t_{N}}$, where ${\mathrm{SOC}}_{t_{0}}$ and ${\mathrm{SOC}}_{t_{N}}$ are the SOC at the beginning and end of the operating cycle.

### 4.3. Solving Model Based on DQN

In the two-layer optimization model, the inner layer optimization and the outer layer optimization belong to different optimization problems. Therefore, this paper uses different decision-making methods for the inner model and the outer model to improve the efficiency of energy storage configuration optimization.

Considering that the single charging loss in the objective function of the inner layer model is a nonlinear term, a linear transformation method is used to construct a typical 0–1 mixed integer linear programming (MILP) problem [36]. This problem can be modeled using the Yalmip toolbox based on MATLAB, and the Gurobi solver is invoked to solve it [37]. Since the outer optimization model is a typical single-objective nonlinear optimization problem, the MATLAB-based genetic algorithm toolkit gatbx is proposed to solve [38]. However, the genetic algorithm has no mechanism for timely feedback information and the speed of search and training is relatively slow. At the same time, it is difficult for this algorithm to deal with nonlinear constraints well.

Reinforcement learning (RL) is an area of machine learning inspired by behavioral psychology, concerned with how software agents ought to take actions in an environment so as to maximize some notion of cumulative reward. It uses interactive trial-and-error learning without preprocessing a large amount of labeled data, which can well realize the modeling of optimization problems. RL has great decision-making abilities, but it does not deal with perception problems enough. The RL framework is shown in Figure 4.

The action value function $Q_{\pi }\left(s,a\right)$ represents the execution of action *a* in the current state *s*, and loops to the end of learning according to the strategy *π*. It can be expressed as
(21)$$Q^{\pi }\left(s,a\right)=E_{\pi }\left[\sum_{k=0}^{n}\delta^{k}r_{t+k+1}|s_{t}=s,a_{t}=a\right].$$
where 0 ≤ *δ* ≤ 1 is the discount coefficient, which can reduce the impact of future rewards on the current action. The action value follows the Bellman optimal equation, which can be expressed as
*Q^π^*(*s*, *a*) = *E**_s_*_′~*s*_[*r* + *δ*max*_a_*_′_*Q*(*s*′, *a*′)|*s*, *a*].(22)

Deep reinforcement learning (DRL) combines the perception ability of deep learning with the decision-making ability of RL, and provides a solution to the perceptual decision-making problem of complex systems. The *Q* value function can be solved by an iteration operation based on the Bellman equation. In practical applications, the approximate estimation of *Q* value function can be achieved based on DNN and linear function, which further improves the application performance of DRL.

A deep Q-network (DQN) is a DRL that combines deep learning and Q-learning with good convergence performance. DQN architecture consists of a current value network, target value network, error function, playback memory unit, and so on. It utilizes a DNN to estimate the action value function. To address the instability and non-convergence of approximating an action value function using DNN, experience replay playback mechanism and the target network are used to solve these disadvantages. A lithium battery charging and discharging strategy based on an improved deep Q-network (IDQN) is proposed, which achieves a good performance [39]. IDQN based on TD-Error (temporal difference error) and modified ReLU (M-ReLU) function can improve the learning efficiency of a Q-network. At the same time, the M-ReLU function can alleviate the problem of neuronal death and has a better robustness to input changes. The reward function only considers charging/discharging action, and the mathematical expressions are described as
(23)$$r_{t}=\left\{\begin{array}{c} \left(price\left(t\right)-price_{\min}\right)\times p_{ba\_ch}\times T\\ \left(price_{\max}-price\left(t\right)\right)\times p_{ba\_dis}\times T \end{array}\right.,$$
where *T* is the length of the charging/discharging period. *price*(*t*) is the electricity price at time *t*. $p_{ba\_ch}$ and $p_{ba\_dis}$ are the charging power and discharging power of battery, respectively. *price_*_max_ and *price_*_min_ are the highest electricity price and lowest electricity price, respectively. However, the reward function does not consider the impact of battery life loss, which restricts the performance of the energy storage configuration optimization.

Considering the complexity of various energy storage configurations in microgrids, in order to improve the application efficiency of energy storage, we propose using IDQN to solve the outer layer optimization model. In order to improve the accuracy of the outer optimization model, the reward function of the agent can be described as
(24)$$r_{t}=-c\left(t\right)-\lambda \left[1-I\left(\mathrm{SOH}\left(t\right)\geq 0.2\right)\right],$$
where *I*(*x*) is the indicator function. The value of *I* is 1 when *x* is true, otherwise *I* is 0. SOH(*t*) ≥ 0.2 is the out-of-bounds penalty. *λ* is the penalty factor, which is a very large number. Actions that violate SOH constraints will be introduced into the reward design.

State *s_t_* should contain enough information for decision-making, including the battery investment cost, battery replacement cost, equipment operation and maintenance cost, fuel cost, annual power purchase cost, capital recovery factor, and SOH. In order to simplify the complexity of the model, several groups of charging and discharging actions $a_{t}\in \left\{-1,-0.9,-0.95,0,0.9,0.95,1\right\}$ are set [39]. Algorithm 1 describes the specific steps of the two-layer optimization model. At time *t*, the agent observes the state of the environment, selects an outer action according to its policy, and inputs the outer model action to the inner model. The inner layer model uses the MILP solver to find the best inner layer action. All actions act on the environment to get the next state and store the experience into the experience pool for DQN training.
**Algorithm 1**. Two-layer optimization model.1: Randomly initialize Q-network *Q*(*s*, *a*; *θ*) with weights *θ*;2: Initialize *l* = 360 days, *t* = 1;3: **for** episode = 1, *M*4:   Receive initial state *s*_1_;5:   **for**
*t* = 1, *l*6:      With probability *ε* select a random action *a_t_*;7:      *r* = getRandom();8:      **if**
*r* < *ε*
**then**9:       Selet *a_t_* randomly;10:      **else**11:      Select *a_t_* = *argmax_a_Q*(*s_t_*, *a_t_*; *θ*);12:      **end if**13:     *ε* = *ε* − △*ε*14:      Solve the MILP problem and get the optimal *c*(*t*) and a^t;15:      Execute action *a_t_* and a^t;16:       Get reward *r_t_* and new state *s_t_*_+1_;17:       Store transition (*s_t_*, *a_t_*, *r_t_*, *s_t_*_+1_) in replay memory *D*;18:       Sample random minibatch of transitions from *D*;19:       Calculate accumulative reward by target Q-network with parameters *θ*;20:      Perform a gradient decent learning on Q-network with parameters *θ*;21:    **end for**22: **end for**

## 5. Experiment and Analysis

As shown in Figure 1, the microgrid includes 600 kW photovoltaic, 250 kW CHP, 125 kW heat pump, and 2000 kW/h lithium battery capacity. The load type only considers electrical and thermal loads. Table 2 shows the parameters of the lithium battery [40]. Table 3 shows the parameters of other devices. The price of natural gas is set at 2.5 CNY/m^3^. Figure 5 shows the time-of-use price curve [41]. Figure 6 and Figure 7 show the electrical load and heat load under different states of the microgrid, respectively. Power transmission to the public grid is capped at 1000 kW.

The microgrid planning period is 20 years and the discount rate is 0.06. The learning rate, reward discount rate, training data storage memory, batch size, and step size of the IDQN are set as 0.0005, 0.95, 10,000, 32, and 400, respectively. The step size of 400 indicates that the target network parameters are updated every 400 steps. At the initial moment of training, the value of the ε-greedy algorithm *ε* is set to 1, and a decreasing exploration is performed in steps of 0.0005 until the value of *ε* becomes 0. In order to become closer to the actual situation, this paper selects one day each for working days, peak days, and rest days from each month. The number of three typical days is set to 20, 3, and 8, respectively. The annual load data and photovoltaic output data in the microgrid are from the literature [42].

To verify the effectiveness of the proposed method, this paper analyzes the impact of battery life loss estimation on battery capacity configuration. The battery capacity configuration results and economic parameters of the proposed method are shown in Table 4. Case-1 is a model that does not consider battery life loss. Case-2 is an optimization model for predicting battery life based on SOC [39]. Case-3 is an optimization model based on SOC and IDQN. Case-4 is the proposed scheme based on SOH and IDQN in this paper. Figure 8 shows the battery SOC curves of the different models in January and July.

According to the experimental results, the battery configuration capacity in Case-1 is 1107.45 kW/h. Case-1 does not consider the impact of battery life loss, resulting in higher battery capacity and replacement costs. Case-2 optimizes the configuration strategy of energy storage batteries based on the battery degradation model. Its performance is affected by the battery degradation model. Case-3 uses a DQN to improve the outer optimization model of Case-2 and achieves good performance. In Case-4, the activation function of Case-3 is adjusted based on the actual operating conditions, which further improves the performance of the energy storage battery configuration. After optimization by combining battery life loss, the battery configuration in Case-4 has a reduced capacity of 741.27 kW/h. The maintenance cost of Case-1 is CNY 1.5458 million, which is better than Case-2, Case-3, and Case-4. However, it does not take into account the loss of battery life, and the replacement cost of the battery is as high as CNY 620,900, which ultimately makes the equivalent annual cost the highest. By comparing the results of Case-1, Case-2, and Case-3, the proposed method has a better economy. As shown in Figure 8, the battery SOC curve of Case-4 is smoother than other models, which avoids the rapid charging and discharging of the battery. In Figure 9, the battery decay rate of Case-4 is slower than that of Case-1, Case-2, and Case-3. The life of the battery can be extended. It can be seen from Table 4 that the battery charging/discharging times of Case-4 are 38.16%, 4.72%, and 3.97% higher than those of Case-1, Case-2, and Case-3, respectively.

This paper proposes a double-layer optimal configuration model of electric/thermal hybrid energy storage considering battery life loss, evaluates the investment benefit of energy storage, and reduces the configuration capacity of energy storage batteries. By analyzing the impact of battery life loss on the configuration capacity, the output range of the battery is made smoother, which effectively improves the economy of microgrid operation. At the same time, the electric/thermal hybrid energy storage configuration is conducive to delaying the decay rate of battery life and can better adapt to the complementary characteristics of multiple energy sources.

Based on the above analysis, the proposed method increases the maintenance cost but effectively prolongs the service life of the battery. Therefore, the optimal configuration of the battery capacity greatly reduces the replacement cost of the battery and takes into account both the battery life and the system operating economy.

## 6. Conclusions

In this paper, a quantification model of battery life loss based on SOH is constructed based on deep learning technology. At the same time, by analyzing the multi-energy complementary scenarios of the microgrid, a two-layer optimal configuration model of energy storage considering battery life loss is designed, which solves the problems faced by energy storage planning and economic operation. Compared with different energy storage configuration schemes, the proposed scheme can take into account the economy and battery life and improve the operation performance of the microgrid. The method proposed in this paper does not consider the impact of new energy sources and loads. Therefore, in the next step, we will consider the impact of various new formats on the configuration of electric/thermal hybrid energy storage in the new power system environment to improve the robustness of the optimization method.

## Figures and Tables

**Figure 1 entropy-24-00630-f001:**
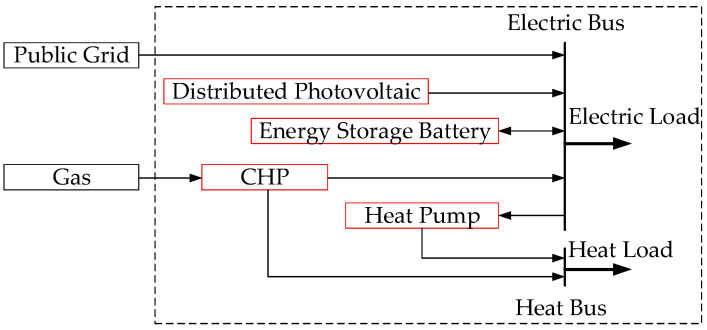
The structure of microgrid.

**Figure 2 entropy-24-00630-f002:**
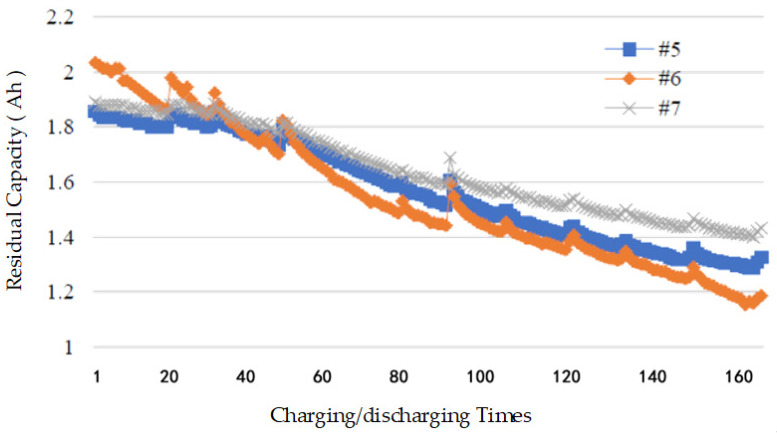
The relationship between the residual capacity and the charging/discharging times.

**Figure 3 entropy-24-00630-f003:**
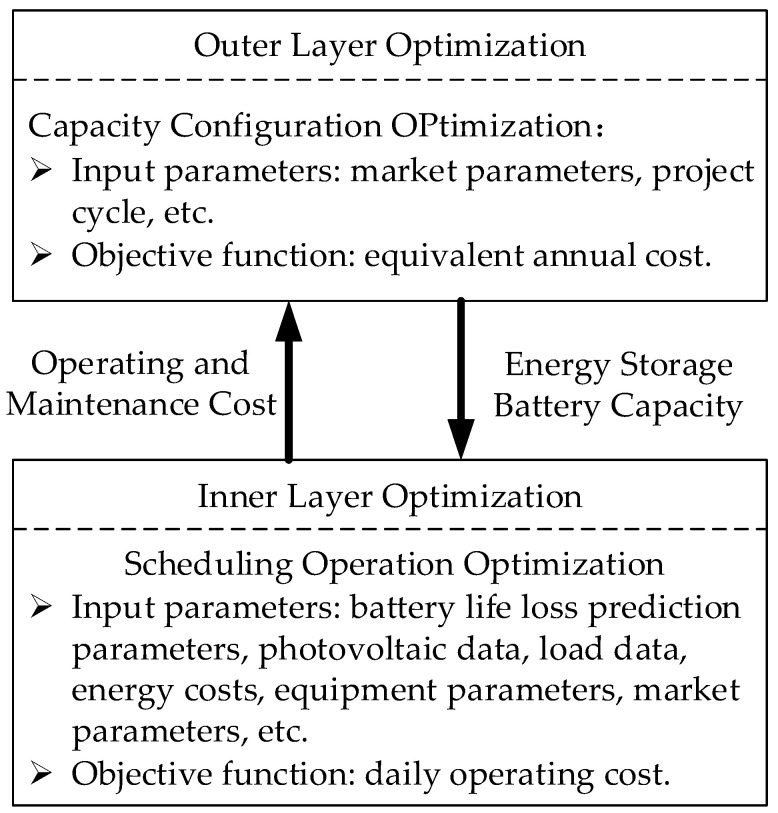
Two-layer optimized structure.

**Figure 4 entropy-24-00630-f004:**
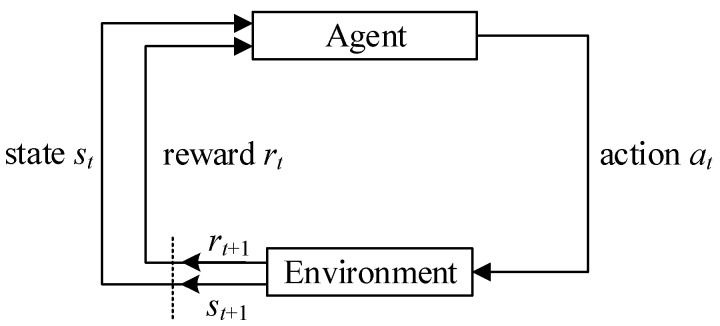
The RL framework.

**Figure 5 entropy-24-00630-f005:**
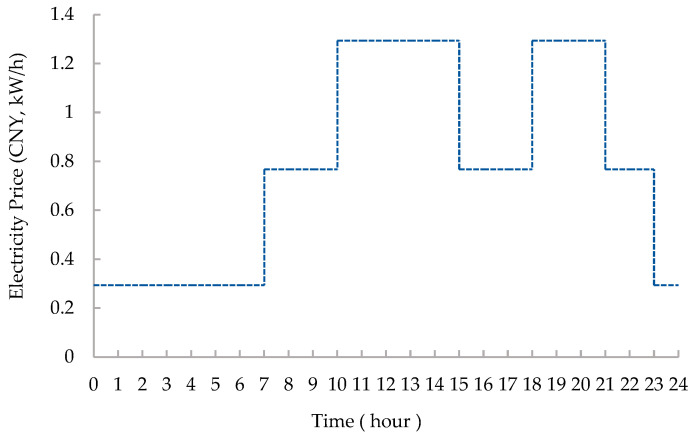
Time-of-use electricity price.

**Figure 6 entropy-24-00630-f006:**
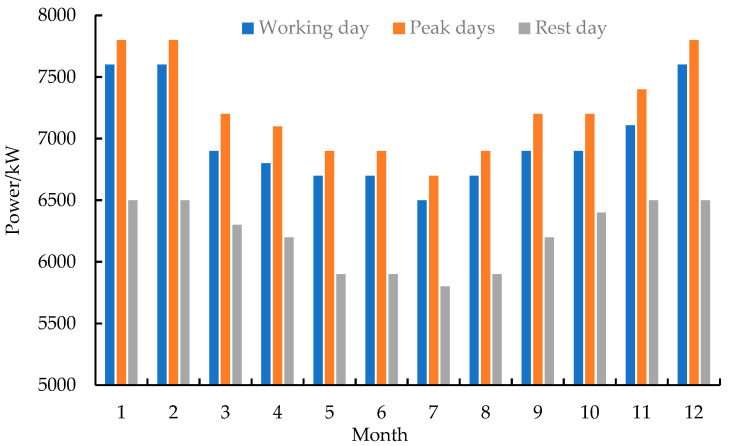
Electrical load.

**Figure 7 entropy-24-00630-f007:**
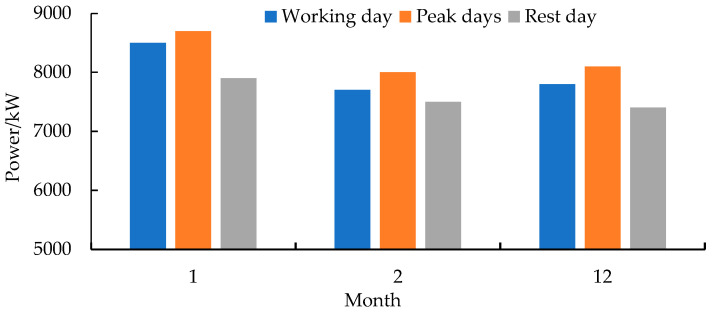
Heat load.

**Figure 8 entropy-24-00630-f008:**
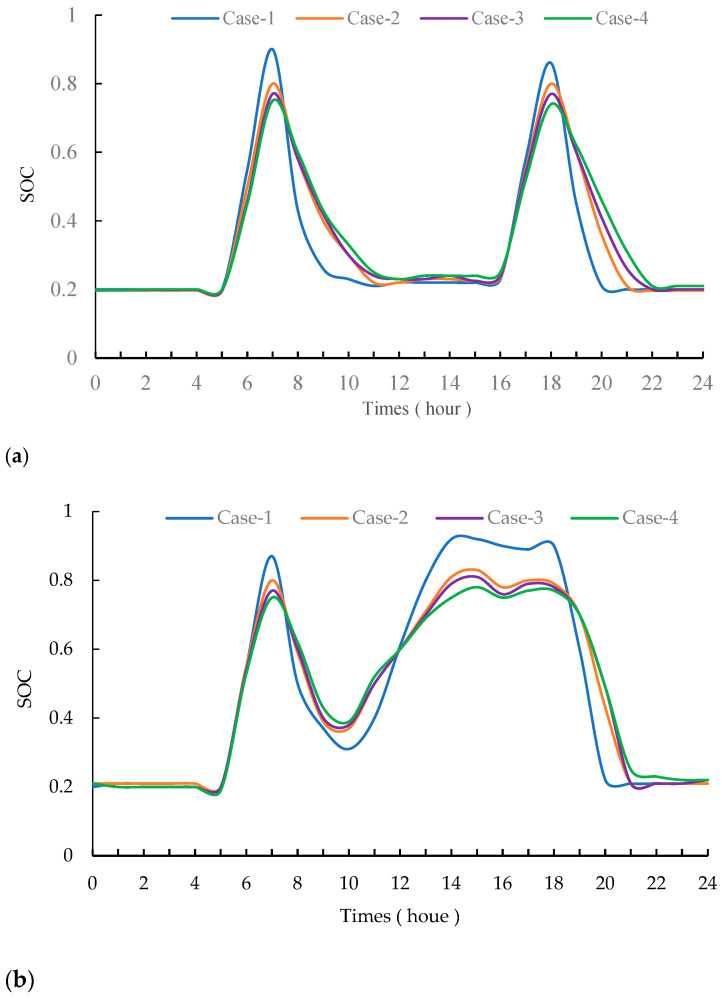
SOC curves of battery on some typical days. (**a**) Peak day in January; (**b**) Peak day in July.

**Figure 9 entropy-24-00630-f009:**
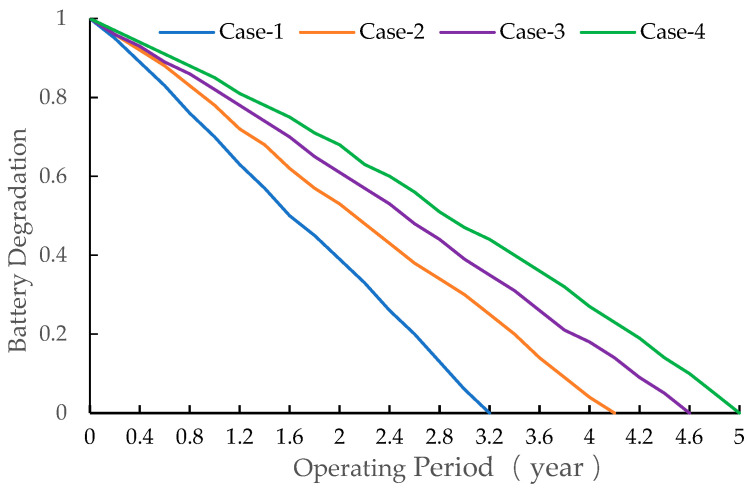
Curves of battery degradation.

**Table 1 entropy-24-00630-t001:** Structure of MobileNetV3-Small-M.

Input	Operator	Output Channels	SE
3 × 300 × 1	conv2d, 3 × 3	16	-
2 × 150 × 16	bneck, 3 × 3	16	√
1 × 75 × 16	bneck, 3 × 3	24	-
1 × 38 × 24	bneck, 3 × 3	24	-
1 × 38 × 24	bneck, 5 × 5	40	√
1 × 19 × 40	bneck, 5 × 5	40	√
1 × 19 × 40	bneck, 5 × 5	40	√
1 × 19 × 40	bneck, 5 × 5	48	√
1 × 19 × 48	bneck, 5 × 5	48	√
1 × 19 × 48	bneck, 5 × 5	96	√
1 × 10 × 96	bneck, 5 × 5	96	√
1 × 10 × 96	bneck, 5 × 5	96	√
1 × 10 × 96	conv2d, 1 × 1	576	√
1 × 5 × 576	pool, 1 × 1	-	-
1 × 5 × 576	conv2d, 1 × 1, NBN	1024	-
1 × 2 × 1024	conv2d, 1 × 1, NBN	1	-

**Table 2 entropy-24-00630-t002:** The parameters of the lithium battery.

Parameter	Value
Self-discharge Rate	0.001
Charging/discharging Efficiency	0.95
Capitalized Cost (CNY, kW/h)	1500
Maintenance Cost (CNY, kW/h)	0.026

**Table 3 entropy-24-00630-t003:** The parameters of other equipment.

Type	Parameter	Value
CHP	Gas-to-electric Ratio	0.3
	Heat-to-electric Ratio	1.36
	Maintenance Cost (CNY, kW/h)	0.05
Heat Pump	Energy Efficiency Ratio	3.8
	Maintenance Cost (CNY, kW/h)	0.026
Photovoltaic	Maintenance Cost (CNY, kW/h)	0.025

**Table 4 entropy-24-00630-t004:** Configuration results and economic parameters (kW/h, CNY).

No.	Battery Capacity	Equivalent Annual Cost	Replacement Cost	Maintenance Cost	Charging/Discharging Times
Case-1	1107.45	216.67	62.09	154.58	739
Case-2	773.84	194.24	30.93	163.31	975
Case-3	756.29	192.08	26.4	165.68	982
Case-4	741.27	189.11	22.21	166.87	1021

## Data Availability

Data will be provided upon request.

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
