# Peer review of "Optimal Energy-Storage Configuration for Microgrids Based on SOH Estimation and Deep Q-Network"

_entropy, 2022, doi:10.3390/e24050630_

Round 1

Reviewer 1 Report

- The novelty of this study is not clear and should be explicitly stated in the introduction.

- At the end of the introduction clearly outline the objectives of the paper in bullet point form.

- I would advise adding colour coding to improve Figure 1.

- In Section 3, why was the battery data from NASA used i.e. why those batteries in particular? Also why were the 5th 6th and 7th batteries chosen? It seems quite an arbitrary selection.

- The x-axis in Figure 2 contains too many numbers. Increments of 20 would be sufficient.

- In Section 3.1, the MobileNetV1 and MobileNetV2 models are mentioned without any background information. It would be beneficial to include a brief sentence on both, what they were used for etc.

- The equations in Section 3.1 should be numbered, and the parameters x and h need to be identified.

- All equations throughout the paper should be numbered.

- In Section 5, you state "As shown in Figure 1, the microgrid includes 600 kW photovoltaic, 250 kW CHP, 125 kW heat pump and 2000 kW/h lithium battery capacity". Figure 1 does not contain this information.

- Did you consider using an algorithm other than MILP in the optimization process? Which cognate studies have shown this to be an effective method for solving these types of problems?

- In Section 5, Case 1,2,3,4 should be explained in much greater detail - what are the loads involved and what is the relevance of the microgrid for each case?

- The Analysis is very brief; it should include more critical analysis of the results for each of the four cases and comparisons of the results in this paper to cognate studies.

Reviewer 2 Report

This paper presents the optimal energy storage configuration. The previous works are well reviewed. The proposed method is well described.

  1. Equation numbers should be added.
  2. The microgrid information such as distribution line and load capacity should be added.
  3. How do you derive the values in Table 4?

Reviewer 3 Report

The article is suitable for the aim and scope of the Journal.

Scientific literature mentioned in this study is limited and not very updated.

Please check last publications in highly reputed Journals from top-ranked studies. See works from Prof. Mazzoni at NTU Singapore, prof. Marechal at EPFL and other eminent scientists.

Stress the novelty of your contribution compared to the recent literature.

Check English spell.

Expand the discussion of the results.

Round 2

Reviewer 3 Report

The authors improved the manuscript by implementing the suggested changes.